

# Sensitivities of Amazonian clouds to aerosols and updraft speed

Micael A. Cecchini[1], Luiz A. T. Machado[1], Meinrat O. Andreae[2,12], Scot T. Martin[3], Rachel I. Albrecht[4], Paulo Artaxo[5], Henrique M. J. Barbosa[5], Stephan Borrmann[2,6], Daniel Fütterer[7], Tina Jurkat[7], Christoph Mahnke[2,6], Andreas Minikin[8], Sergej Molleker[6], Mira L. Pöhlker[2], Ulrich Pöschl[2], Daniel Rosenfeld[9], Christiane Voigt[6,7], Bernadett Weinzierl[7,10], Manfred Wendisch[11]

[1]Centro de Previsão de Tempo e Estudos Climáticos, Instituto Nacional de Pesquisas Espaciais, Cachoeira Paulista, Brasil.
[2]Biogeochemistry, Multiphase Chemistry, and Particle Chemistry Departments, Max Planck Institute for Chemistry, P.O. Box 3060, 55020, Mainz, Germany.
[3]School of Engineering and Applied Sciences and Department of Earth and Planetary Sciences, Harvard University, Cambridge, Massachusetts, USA.
[4]Departamento de Ciências Atmosféricas, Instituto de Astronomia, Geofísica e Ciências Atmosféricas (IAG), Universidade de São Paulo (USP), Brasil.
[5]Instituto de Física (IF), Universidade de São Paulo (USP), São Paulo, Brasil.
[6]Institut für Physik der Atmosphäre (IPA), Johannes Gutenberg-Universität, Mainz, Deutschland.
[7]Institut für Physik der Atmosphäre, Deutsches Zentrum für Luft- und Raumfahrt (DLR), Oberpfaffenhofen, 82234 Wessling, Deutschland.
[8]Flugexperimente, Deutsches Zentrum für Luft- und Raumfahrt (DLR), Oberpfaffenhofen, Deutschland.
[9]Institute of Earth Sciences, The Hebrew University of Jerusalem, Israel.
[10]Faculty of Physics, University of Vienna, Boltzmanngasse 5, 1090 Wien, Austria.
[11]Leipziger Institut für Meteorologie (LIM), Universität Leipzig, Stephanstr. 3, 04103 Leipzig, Deutschland.
[12] Scripps Institution of Oceanography. University of California San Diego, La Jolla, CA92093, USA.

*Correspondence to*: M. A. Cecchini (micael.cecchini@cptec.inpe.br)

**Abstract.** The effects of aerosol particles and updraft speed on warm-phase cloud microphysical properties are studied in the Amazon region as part of the ACRIDICON-CHUVA experiment. Here we expand the sensitivity analysis usually found in the literature by concomitantly considering cloud evolution, putting the sensitivity quantifications into perspective in relation to in-cloud processing, and by considering the effects on droplet size distribution (DSD) shape. Our in-situ aircraft measurements over the Amazon basin cover a wide range of particle concentration and thermodynamic conditions, from the pristine regions over coastal and forested areas to the highly biomass-burning-polluted southern Amazon. The quantitative results show that particle concentration is the primary driver for the vertical profiles of effective diameter and droplet concentration in the warm phase of Amazonian convective clouds, while updraft speeds have a modulating role in the latter and in total condensed water. The cloud microphysical properties were found to be highly variable with altitude above cloud base, which we used as a proxy for cloud evolution since it is a measure of the time droplets were subject to cloud processing. We show that DSD shape is crucial in understanding cloud sensitivities. The aerosol effect on DSD shape was found to vary with altitude, which can help models to better constrain the indirect aerosol effect on climate.



# 1 Introduction

The Amazon basin can serve as a natural laboratory to study anthropogenic effects on cloud microphysical and radiative properties. In its remote parts, an absence of pollution similar to the pre-industrial era still prevail, while in other regions, cities and biomass burning emit high amounts of aerosol particles into the atmosphere. This is especially important during the dry season, when rainout is less frequent (Artaxo et al., 2002; Kuhn et al., 2010; Martin et al., 2010). Under background conditions, cloud condensation nuclei (CCN) consist mostly of secondary organic aerosol (SOA) particles formed by the oxidation of volatile organic compounds (VOCs), which condense and grow sufficiently to form CCN (Pöschl et al., 2010). Anthropogenic emissions may enhance the oxidation process, leading to increased SOA and CCN concentrations (Kanakidou et al., 2000; Hallquist et al., 2009). Even though aerosol particles can be scavenged by precipitation, nano-particles produced in the upper troposphere can be transported downwards by deep convective systems, approximately reestablishing the surface aerosol concentration (Wang et al., 2016).

These processes illustrate the complex feedbacks between the vegetation, the aerosols serving as CCN, and the clouds providing water to the vegetation. There are, however, still plenty of open questions. The main difficulty in this regard is the quantitative comparison of the aerosol effect to other processes, given that the anthropogenic influences alter more than just aerosol particle concentrations. Human activities associated with urbanization and agriculture change the local landscape and the Earth's surface properties, also altering the energy budget (Fisch et al., 2004) and consequently the thermodynamic conditions for cloud formation. According to Fisch et al. (2004), the convective boundary layer is deeper over pasture during the dry season because of the increased sensible heat fluxes. This effect results in greater cloud base heights with potentially stronger updrafts, which should also be considered when analyzing the aerosol effect.

One possible way to compare different effects on cloud microphysical properties is through a sensitivity calculation. It can provide specific quantifications of aerosol and thermodynamic effects on cloud microphysical quantities. One such sensitivity study was proposed by Feingold (2003), where the author calculates cloud droplet number concentration ($N_d$) sensitivities to several aerosol and thermodynamic drivers, such as total aerosol particle concentration ($N_a$), updraft speed ($w$), and liquid water content ($LWC$). However, this analysis was limited to adiabatic stratocumulus clouds where collision-coalescence was not considered. Another modeling study was proposed by Reutter et al. (2009), where they identified three regimes that modulate the $N_d$ sensitivity. The regimes are aerosol-limited, updraft-limited, and the transition between them. The authors highlight that the $N_d$ dependence on $N_a$ and $w$ may vary given their relative magnitudes. This study is limited to cloud base, therefore not addressing cloud evolution. The Reutter et al. (2009) study was extended by Chang et al. (2015), who took into account the evolution of the systems by considering the sensitivities on precipitation and ice phase, but was relatively limited in terms of representativeness because of the use of a 2D model. Satellite studies (e.g., Bréon 2002; Quaas et al. 2004; Bulgin et al. 2008) have an intrinsic limitation given the characteristics of the remote sensors. This kind of study usually deals with vertically-integrated quantities and frequently focuses on oceanic regions because of the favorable surface contrast.





The main goal of this study is to expand the sensitivity calculations usually found in the literature to include: 1) aerosol and thermodynamic effects on cloud droplet number concentration, size, and shape of the size distribution; 2) comparison with the effect of cloud evolution, i.e., in-cloud processing; and 3) in-situ observations of the less frequently studied convective clouds over tropical continental regions. For this purpose, we report on recent measurements over the Amazon rainforest during the

5 ACRIDICON-CHUVA campaign (Wendisch et al. 2016), where a wide variety (in terms of aerosol concentrations and thermodynamic conditions) of cloud types were probed. We quantify the aerosol-induced changes in cloud microphysical properties and compare them to the effects of updraft intensity, which are related to thermodynamic properties, over different regions in the Amazon. Both processes are analyzed with a focus on cloud evolution. Our methodology should prove useful for better understanding aerosol-cloud interactions over the Amazon, which is a region, as are the tropics as a whole, with poor

forecasting skill (Kidd et al., 2013). Section 2 describes the experiment, its data, and the methods used for the analysis. Results are presented in Section 3, followed by the conclusions in Section 4.

## 2 Methodology

### 2.1 Campaign and methodology

During the years 2014 and 2015, the GoAmazon2014/5 campaign took place in the Amazon to improve our understanding

regarding aerosol particles, atmospheric chemistry, clouds, radiation, and their interactions (Martin et al., 2016). In conjunction with the second Intensive Operations Period (IOP2) of this experiment, the ACRIDICON-CHUVA campaign took place during September-October 2014 (Wendisch et al., 2016). It included 14 research flights with the German HALO (High Altitude and Long Range Research Aircraft). A previous campaign dedicated to study aerosol-cloud interactions took place in the Amazon in 2002 (LBA-SMOCC; Andreae et al., 2004), but it had been relatively limited in terms of range and ceiling of the aircraft

measurements. The high endurance of the HALO aircraft, which carried sophisticated microphysical, aerosol and solar radiation instrumentation, allowed for long-range flights from remote areas in northern Amazon, to the deforestation arc in the south and to the Atlantic coast in the eastern part (Figure 1). The flights were planned to cover five different mission types focusing on different cloud, aerosol, chemistry and radiation processes (see Wendisch et al., 2016 for details). The flights were numbered chronologically as ACXX, where XX varies from 07 to 20. For this study, the cloud profiling missions are of

particular interest and their respective locations are shown in Figure 1. In this study, we take advantage of HALO's capabilities to compare different types of clouds formed over different Amazonian regions, focusing on their warm microphysics. In addition to the HALO measurements, ground-based equipment was also operated in and near Manaus city (Machado et al., 2014; Martin et al., 2016).

The results shown here were obtained from the measurements of four different instruments (for a list of all HALO instruments,

see Wendisch et al., 2016), covering aerosol, cloud and meteorological properties. We will focus on aerosol and CCN number concentrations, cloud droplet size distributions (DSD), and updraft speed. The instruments are briefly described below.



### 2.1.1 CCP

For the cloud droplet size distribution measurements, a modified Cloud Combination Probe (CCP, manufactured by Droplet Measurement Technologies, Inc., Boulder, CO, USA) was adopted on HALO covering an overall size diameter range from 3 µm to 950 µm. The probe consists of two separate instruments, the CDP (Cloud Droplet Probe; Lance et al., 2010; Molleker

et al., 2014) and a grayscale optical array imaging probe (CIPgs, Cloud Imaging Probe, Korolev, 2007). By means of a two-dimensional shadowcast technique the CIPgs detects cloud particles with size diameters ranging from 15 µm to 2000 µm. The in-house developed analysis algorithm from MPI and IPA in Mainz sizes and sorts the recorded images into bins of roughly 15 µm bin width in dependency on particle shapes and dimensions. The CDP is an optical particle counter detecting scattered laser light (in forward direction) arising from individual particles passing through the illuminated optical sample area (Lance

et al., 2010; Molleker et al., 2014). The optical sample area has a cross section of 0.2 mm$^2$ ($\pm$ 15%) perpendicular to the flight direction. The CDP detects particles with sizes from 3 µm to 50 µm, and classifies these into size histograms of bin widths between 1 and 2 µm. In addition to size histograms recorded at 1Hz frequency, the CDP stores single particle data (signal amplitude and µs-resolved detection time) of continuous intervals with up to 256 particles every second. This feature can be used to assess the spatial distribution of the droplets in case of multi modal size distributions (Klingebiel et al., 2014). The

main uncertainties for the CCP size distributions are due to the uncertainty of the sample area (and thus the scanned air volume), as well as counting statistics. We applied a filter to eliminate DSDs with concentrations lower than 1 cm$^{-3}$ for D<50 µm or lower than 0.1 cm$^{-3}$ for D>50 µm.

### 2.1.2 AMETYST-CPC

The aerosol concentrations used in this study refer to the total concentration of particles measured with a butanol-based

condensation particle counter (CPC). Four CPCs were deployed on HALO as part of the new basic aerosol instrument package for HALO named AMETYST (Aerosol MeasuremenT sYSTem), described in detail by D. Fütterer (DLR, PhD thesis in preparation) and which also includes two Grimm 1.129 OPC (Optical Particle Counters), a two-channel thermal denuder operated at 250 °C, a Radiance Research 3-wavelength PSAP (Particle Soot Absorption Photometer), and optionally two DMAs (Differential Mobility Analysers). AMETYST is operated behind the HALO sub-micrometer aerosol inlet (HASI). The

CPCs are Grimm 5.410 models, operating at two different flow rates. The CPC internal butanol saturation setting is user-selectable to vary minimum detectable particle sizes. Data used in this study were obtained from 0.6 l/min 5.410 CPC set to a nominal lower cut-off size of 10 nm. Concentrations reported are normalized to standard temperature and pressure conditions. Original data are recorded at 1 Hz temporal resolution. In-cloud data at altitudes below 9 km were removed from the dataset based on cloud probe data (here CAS-DPOL instrument of DLR) to exclude apparent sampling artefacts of the inlet in the

presence of liquid droplets in clouds.





### 2.1.3 CCNC

A Cloud Condensation Nuclei Counter (CCNC) was used to obtain CCN number concentrations. The instrument has two columns with continuous flow longitudinal thermal-gradient where the aerosol particles are subject controlled supersaturation (S) conditions. When particles travel longitudinally in the center of each column, they grow by water condensation (depending

on their physical and chemical compositions) and are counted as CCN if they reach 1 μm in size (1 Hz sampling rate). It is manufactured by Droplet Measurement Technologies (DMT) – Roberts and Nenes (2005). Calibrations were performed between flights following Rose et al. (2007). At one column, S was set to be relatively constant at S≈0.55%, while the other was subject to 100-s stepping variations between 0.2% and 0.55%.

### 2.1.4 BAHAMAS

Vertical wind speeds were obtained from the BAsic HALO Measurement And Sensor System (BAHAMAS) sensor installed at the nose of the aircraft (Wendisch et al., 2016). The 3D wind measurements were calibrated following Mallaun et al. (2015), resulting in an uncertainty of 0.3 m s$^{-1}$ for the horizontal and 0.2 m s$^{-1}$ for the vertical components.

### 2.2 Sensitivity calculation

Several earlier studies calculated cloud sensitivity to aerosols and/or updrafts (Feingold, 2003; McFiggans et al, 2006; Kay
and Wood, 2008; Reutter et al, 2009; Sorooshian et al, 2009; Kardys et al, 2012; Chang et al, 2015), but they were usually limited in scope by not considering the factors that contribute to the cloud microphysics individually. This study aims to expand the sensitivity methodology by concurrently considering cloud evolution, updraft speed, and aerosol effects on clouds and by taking advantage of the comprehensive ACRIDICON-CHUVA dataset to represent different kinds of clouds and thermodynamic conditions. As pointed out by Seinfeld et al. (2016), major field campaigns provide a key opportunity for
improving our knowledge of the aerosol-cloud-climate interactions, further motivating the results to be presented here.
Three factors will be considered as the main drivers of cloud microphysical properties, each representative at least partially of thermodynamic and aerosol conditions and cloud evolution. For the aerosol characterization, we will use averaged concentrations measured by the AMETYST-CPC (referred here as $N_a$ - see Table 1) at the cloud base level. This level was obtained from the CCP-CDP measurements as the lowest level where the $LWC$ is higher than 0.01 g m$^{-1}$. As the profiles always
started by cloud base penetrations, this ensures a precise estimation of cloud base altitude. Table 1 also shows that cloud condensation nuclei (CCN) concentrations were proportional to $N_a$ for the chosen instrument supersaturation. A linear fit between the two concentrations results in R$^2$=0.96, with angular and linear coefficients equal to 1.57 and 243 cm$^{-3}$, respectively. For the purposes of the sensitivity calculations, we will use $N_a$ instead of CCN concentrations because they are not dependent on instrument or cloud supersaturations. The sensitivity calculation (see below) uses derivatives of the concentrations, so the
choice of $N_a$ or CCN should have no significant impact on the results to be presented here. The most pristine clouds are observed near the coast (AC19), followed by the ones measured over the forest. The flights AC7, AC12, and AC13 each





showed increasing aerosol concentrations as the flights moved towards the southern Amazon. For the flights closer to Manaus city, the aerosol loading of the clouds depends on localized aspects such as small-scale biomass burning and the pollution plume from urban/industrial activities (Cecchini et al., 2016).

The second factor that affects cloud microphysics is the updraft intensity ($w$). It, along with the aerosol population, defines the

supersaturation inside the clouds and thus affects the droplets condensational growth. The intensity of the updrafts depends both on meteorological conditions (e.g., temperature and humidity profiles) and on the latent heat release of condensing water. Aerosols may indirectly affect the amount of latent heat released (smaller droplets in polluted clouds have favorable area-to-volume ratio), but the speed of the ascending air can be understood as a response to the thermodynamic conditions in the clouds. Therefore, $w$ can be used as a benchmark to compare different clouds subject to similar cloud-microphysics-relevant

thermodynamic conditions.

Lastly, it is important to have an estimate of how cloud microphysical properties evolve throughout the system evolution. More importantly, how to detect similar cloud stages over the different flights for comparison. The HALO cloud profiling missions were planned to capture growing convective clouds in the different Amazonian regions. The aircraft penetrated the systems first at cloud base and then at ascending altitudes in the cloud tops of the growing convective elements. This strategy allows

the use of altitude above cloud base (herein referred as $H$, in meters, also known as cloud depth) as proxy for cloud evolution. Measurements at higher altitudes reflect later stages of the cloud lifecycle as the systems develop upward. We use the derivatives of the microphysical properties with respect to $H$, which can be understood as variations during the cloud evolution. This will put the sensitivities to $N_a$ and $w$ into perspective, highlighting the importance to detect cloud stage.

The sensitivities are calculated as partial derivatives on a natural log scale. In this way, they are normalized for quantitative

comparison. Based on the terminology in the literature (e.g., Feingold 2003; Chang et al, 2015), we consider the sensitivities as follows:

$$S_Y(X_i) = \left.\frac{\partial lnY}{\partial lnX_i}\right|_{X_j, X_k} \tag{1}$$

where $X$ is the independent variable, i.e., $w$, $N_a$ and $H$, and $Y$ is the cloud microphysical property of interest. For the sensitivity calculation, we will focus firstly (Section 3.2) on cloud droplet number concentration ($N_d$) and effective droplet diameter ($D_{eff}$)

of cloud DSD with D<50 μm. In Section 3.3 we also consider the sensitivities in ($LWC$), relative dispersion ($\varepsilon$) and curvature parameter ($\Lambda$, see respective text for details). The three factors chosen for $X$ in this study are not necessarily independent; therefore, in order to follow the partial derivative formalism, we include the criteria expressed by the vertical line (Equation 1). The subscript in $X$ identifies the different independent variables considered. This notation means that two independent variables remain constant while the sensitivity to the third is being calculated. As an example, the sensitivity of $N_d$ to $N_a$, $w$,

and $H$ can be expressed as:

$$S_{N_d}(N_a) = \left.\frac{\partial lnN_d}{\partial lnN_a}\right|_{w,H} , \; S_{N_d}(w) = \left.\frac{\partial lnN_d}{\partial lnw}\right|_{N_a,H} , \; S_{N_d}(H) = \left.\frac{\partial lnN_d}{\partial lnH}\right|_{w,N_a} \tag{2}$$





Equation 2 recognizes that several parameters can affect $N_d$, and they should be analyzed individually. Other sensitivities, such as $S_{D_{eff}}(N_a)$ or $S_{D_{eff}}(w)$, are obtained analogously.

As it is not feasible to analyze the sensitivities under exactly constant conditions as in Equation 2, we decided to use $N_a$, $w$ and $H$ intervals instead. These quantities were binned into {0 cm$^{-3}$, 500 cm$^{-3}$, 1000 cm$^{-3}$, 3000 cm$^{-3}$, 4500 cm$^{-3}$}, {0 m s$^{-1}$, 0.5 m s$^{-1}$, 1 m s$^{-1}$, 2 m s$^{-1}$, 4 m s$^{-1}$, 8 m s$^{-1}$}, and {0 m, 200 m, 500 m, 950 m, 1625 m, 2637 m, 4156 m}, respectively. In this way, there are 4, 5, and 6 $N_a$, $w$, and $H$ intervals, respectively. The values of the bins were chosen in order to maximize the amount of data in each interval, which required growing spacing in $w$ and $H$. We use constant $N_a$ values for each profile and the respective bins effectively group different flights according to the pollution level. Note that flight AC19 falls in the first interval, flights AC9 and AC18 in the second, AC7, AC11, and AC20 in the third and AC12 and AC13 in the fourth (see Table 1). We then produce 4-by-5-by-6 matrices containing averaged $N_d$ and $D_{eff}$ values for the combined intervals, covering all variations possible. By fixing two dimensions and varying the third, we obtain the average variation of the microphysical property to the independent variable of choice. The sensitivity is calculated as linear fits in the natural logarithm scale.

## 3 Results

### 3.1 Cloud droplet size distributions related to different aerosol and thermodynamic conditions

The first qualitative indication of the effect of $N_a$, $w$, and $H$ on cloud microphysical properties can be seen in Figure 2. This figure shows DSDs grouped into four categories according to the aerosol concentration ($N_a$) at cloud base: 1) Maritime clouds, with $N_a \leq 500$ cm$^{-3}$; 2) Clouds under Amazonian background conditions, with 500 cm$^{-3} < N_a \leq 1000$ cm$^{-3}$; 3) Moderately polluted clouds, with 1000 cm$^{-3} < N_a \leq 3000$ cm$^{-3}$; and 4) Polluted clouds, with $N_a > 3000$ cm$^{-3}$. Solid lines in Figure 2 represent DSDs for neutral vertical speed (-1 m s$^{-1} \leq W \leq 1$ m s$^{-1}$) while the DSDs with dashed and dot-dashed lines indicate the up- and downdraft regions, respectively ($|W| > 1$ m s$^{-1}$; note that we use $W$ to differentiate from $w$ which refers only to the updraft portion). They represent averages for all profiles matching the aerosol intervals chosen (1 maritime, 2 Amazonian background conditions, 3 moderated pollution and 2 polluted). Individual profiles can be found in the supplement material (Figures S1-4). It is evident that aerosols and updrafts affect the droplet size distribution and its evolution in different ways and magnitudes. Clouds that develop under similar aerosol conditions tend to have similar DSDs not only at cloud base but also higher in the warm layer. The individual profiles shown in Figures S1-4 confirm the pattern that is evident in Figure 2. On the other hand, the updraft effect is limited to modulations of the DSDs, especially in the D<10 μm range. Note that DSDs subject to similar $w$ can be widely different depending on the respective pollution. The resulting vertical evolution of the clouds is dependent of the $N_a$ value, being more pronounced the cleaner the clouds are. We only observed significant concentrations of precipitation-sized droplets (e.g., > 100 μm) for $N_a < 3000$ cm$^{-3}$.

The main motivation for calculating sensitivities is to quantify and compare the role of $N_a$, $w$, and $H$ in the formation and evolution of the DSDs as seen in Figure 2. In this way, it will be possible to check the magnitudes of the effects of aerosols,



thermodynamics (as seen from the updrafts), and cloud evolution in the determination of the warm-phase characteristics. Note, however, that we are focusing on only one portion of the updraft effects, i.e., the condensation and collision-coalescence effects. For instance, Heymsfield et al. (2009) showed that small droplets carried up by updrafts can significantly participate in the cold processes of the clouds, which are not addressed here. This study considers the first stage of the cumulus clouds

just before the formation of ice particles. Regardless, Figure 2 evidences that all three chosen independent variables have specific roles in determining cloud DSDs characteristics. Together they explain most of the warm-phase properties.

## 3.2 Comparing the main drivers of bulk microphysical properties of Amazonian clouds

For quantitative comparisons, it is interesting to consider bulk DSD properties such as $N_d$ and $D_{eff}$ instead of the whole DSD as in Figure 2. We will quantify the influence of $N_a$, $w$, and $H$ in these properties as a means to understand the effects on the

10 overall DSD. This analysis will be complemented by the study of the DSD shape in next section. By comparing the sensitivities of cloud droplet concentration and size to $N_a$ and $w$, it is possible to make a comparison that represents, at least partially, the contrasts between the importance of aerosols and thermodynamics on cloud characteristics. A significant portion of the previous work in this field was dedicated to understand the processes that lead to the observed $N_d$. Twomey (1959) provides theoretical considerations of $N_a$ and $w$ effects on the supersaturation, which ultimately defines $N_d$ for a given CCN spectra.

More recent studies report on observations and modeling efforts to portray these processes in different regions of the world, calculating cloud sensitivities to both updraft speed and aerosol conditions. By analyzing aerosol and updraft conditions around the globe, Sullivan et al. (2016) note that $w$ can be the primary driver of $N_d$ in some regions. Reutter et al. (2009), using an adiabatic cloud model, argue that $N_d$ sensitivities to aerosol concentrations and $w$ can vary depending on their relative magnitudes. Adiabatic clouds are not highly sensitive to $w$ (at cloud base) when CCN concentrations are low and vice-versa.

Some studies also consider sensitivities in droplet size, such as Feingold (2003). However, cloud evolution is rarely put into perspective representing a limitation of previous studies. In the following, we will show our extended calculations of the sensitivities, where we consider the effects of aerosols, updraft speed, and $H$ on $N_d$ and $D_{eff}$.

Based on Equation 2, it is evident that there exist several values for each sensitivity. As an example, $S_{N_d}(N_a)$ has different values depending on the chosen pair of $\{w, H\}$. However, given the nature of in-situ measurements, individual $S_{N_d}(N_a)$ values

are associated with reduced sample sizes and, therefore, compromise the statistical confidence. In this case, we present averaged values and the respective standard deviation for all $\{w, H\}$ pairs considered, applying the same calculation to the other sensitivities as well. The intervals chosen for $N_a$, $w$, and $H$ imply that those averages are representative of the lower ~4 km of the clouds, with updrafts up to 8 m s$^{-1}$ and aerosol concentrations ranging from 500 to 4500 cm$^{-3}$.

The results of the $N_d$ and $D_{eff}$ averaged sensitivities (Table 2) reflect the patterns observed in Figure 2. The effective diameter

shows strong association to the aerosol concentration and $H$ while being almost independent of $w$. Specifically regarding $N_a$, the sensitivities calculated represent the first step in the parameterization of the aerosol indirect effect for climate models, i.e., its relation to cloud microphysical properties. Multiple studies have focused on this issue from several observational setups





such as satellite/surface remote sensing and in situ measurements. Pandithurai et al. (2012) provide a compilation of this type of calculation (see their Table 2), showing a high variability among the sources. According to Schmidt et al. (2015), the differences are due to not only the measurement setup but also to the region (ocean/land), the types of clouds, and differences in the methodologies. Remote sensing techniques often retrieve vertically integrated quantities at relatively rough horizontal

resolution, which can smooth the results, meaning lower sensitivities. On the other hand, in-situ airborne measurements are closer to the process scale and may result in more accurate estimates of the aerosol effect (Werner et al., 2014). However, the studies reviewed in Pandithurai et al. (2012) and Schmidt et al. (2015) are mostly for stratus or cumulus clouds over ocean. Additionally, measurements of $w$ were either not available or were not differentiated in most of the previous analyzes, while the results are often integrated in altitude or limited to a specific cloud layer (e.g., cloud top in satellite retrievals). Our study

focuses on tropical convection over the Amazon and takes into account both the updraft speed and altitude of the measurements.

The values of the sensitivities with regard to $N_a$ presented here are among the highest reported in literature. They are not far from the theoretical limit of $S_{N_d}(N_a) = 1$ ($N_d \leq N_a$) and $S_{D_{eff}}(N_a) = -0.33$, which is quite common for in-situ airborne studies (Werner et al., 2014). The limit for $D_{eff}$ is an approximation and stems from the relation (if $LWC$ is held constant)

$D_{eff} \propto \left(\frac{LWC}{N_d}\right)^{1/3}$ (e.g., Martin et al., 1994). Given the precautions taken here to isolate the aerosol effects, these values show that Amazonian clouds are highly sensitive to pollution. Human-emitted particles affect not only the DSDs close to cloud base but also over at least the lower 4 km of the warm-phase domain.

The sensitivities to the updraft speed have a distinct behavior when compared to the aerosol effect. Not only does it show lower values overall but it shows different behaviors for $N_d$ and $D_{eff}$. It shows that even strong updrafts are not able to significantly

increase the effective droplet size by enhancing condensation. In fact, this sensitivity oscillates around zero with slightly negative and positive values (see Table S2) and with relatively low $R^2$. This finding is similar to what Berg et al. (2011) observed in Oklahoma City. Close to cloud base, they found a significant relation between $N_d$ and $w$, and a low correlation between $D_{eff}$ and $w$. Here we show that this feature is not limited to cloud base but persists with altitude on average. Feingold et al. (2003), using an adiabatic cloud parcel model, found a negative value for $S_{D_{eff}}(w)$, with a higher absolute value for

polluted clouds. The result could be explained by activation of smaller aerosol particles with increasing updraft speed, leading to higher concentrations of small droplets that skewed the mean diameter to lower values. Although we observed slightly negative sensitivity for highly polluted clouds at their base (Table S2), our measurements show that the overall averaged $D_{eff}$ is nearly independent of $w$ for the Amazonian clouds.

Freud et al. (2011) and Freud and Rosenfeld (2012) showed similar observations in the Amazon, India, California, and Israel.

They provide theoretical formulations that support some of those observations. These authors showed that the vertical evolution of $D_{eff}$ behaves almost adiabatically because of the predominance of inhomogeneous mixing in convective clouds. In this way, droplet effective size can be obtained from cloud base $N_d$, pressure, and temperature. In fact, this is the framework for a new technique developed to obtain CCN retrievals from satellites (Rosenfeld et al., 2016). Our study provides a new look





at those observations and theoretical considerations by specifically quantifying, without any adiabatic assumption, each process with our formulation of sensitivity.

Comparisons of the sensitivities to $w$ and $N_a$ can be used to infer the roles of the aerosols and thermodynamic conditions on the DSD characteristics. Not only do the aerosols primarily determine the size of the droplets but they also have the biggest impact on the number concentration, high variability in $S_{N_d}(w)$ notwithstanding. This result shows that in terms of the warm layer aerosols play a primary role in determining DSD characteristics.

The sensitivities to H are calculated in order to put the aerosol and updraft effects into perspective regarding cloud evolution. This calculation shows that, on average, droplet growth with cloud evolution is comparable in absolute value and opposite to the aerosol effect. For this reason, studies should take into account the altitude of the measurements. Polluted Amazonian clouds show slower droplet growth with altitude (Cecchini et al., 2016) and $S_{D_{eff}}(H)$ may vary with $N_a$. With lower $S_{D_{eff}}(H)$, $S_{D_{eff}}(N_a)$ possibly increase with altitude. The most important factor evident in Table 2 for $D_{eff}$ is that it shows strong relations with $N_a$ and $H$, while being independent of $w$. This result is of great value for parameterizations or other analyses of cloud droplet size.

Whereas $D_{eff}$ shows a clear relation to $N_a$ and $H$, being relatively constant at fixed altitude, $N_d$ displays a highly variable behavior. The averaged $S_{N_d}(H)$ has a slightly negative value with high standard deviation. There can be either droplet depletion or production with altitude, but the former prevails on average. New droplet activation should be expected in polluted clouds, where not all aerosols are activated at cloud base. In fact, Table S6 shows that $S_{N_d}(H)$ is positive for the most polluted clouds probed when updraft speeds are > 0.5 m s[-1], although $R^2$ values are quite low. Droplet depletion with altitude can be a result of evaporation and/or collection growth. Cecchini et al. (2016) showed that Amazonian background clouds present rather effective collision-coalescence growth, which would suggest a negative $S_{N_d}(H)$ for those clouds. This mechanism is difficult to observe in the present study, with relatively low $R^2$ in the individual $S_{N_d}(H)$ (Table S6). Overall, the highly variable relation between $N_d$ and $H$ suggest that droplet concentration is not closely tied to altitude above cloud base, as it is the case for $D_{eff}$. On the other hand, droplet concentration has significant horizontal variation given different mixture and $w$ conditions, while the effective diameter remains similar at constant altitude levels.

## 3.3 Effects on DSD shape and relation between sensitivities

The use of a parametric function to represent the DSDs can be of interest in order to understand the sensitivities in the overall shape of the DSDs. One function widely adopted in many applications and especially in models (Khain et al. 2015) is the Gamma function. One of the forms of the Gamma function represents the DSDs as:

$$N(D) = N_0 D^\mu \exp(-\Lambda D) \tag{3}$$

where $N_0$, $\mu$ and $\Lambda$ are intercept, shape and curvature parameters, respectively. The advantage of using this function is that it can be analytically integrated, providing relatively simple equations for the DSD parameters. $N_d$, $D_{eff}$, and $LWC$ can be calculated from the moments of the Gamma DSD (units are cm[-3], μm, and g m[-3], respectively):





$$N_d = M_0 \qquad (4)$$

$$D_{eff} = \frac{M_3}{M_2} \qquad (5)$$

$$LWC = 10^{-9} \rho_w \frac{\pi}{6} M_3 \qquad (6)$$

Where $\rho_w$ is the density of liquid water (considered as 1000 kg m$^{-3}$ here) and $M_p$ is the pth moment of the DSD, given by:

$$M_p = \int_0^\infty D^p N(D) dD = N_0 \frac{\Gamma(\mu+p+1)}{\Lambda^{\mu+p+1}} \qquad (7)$$

By substituting (7) into (5) it is possible to write $D_{eff}$ as a function of $N_d$ and $LWC$:

$$D_{eff} = 10^9 \frac{6}{\pi\rho_w} \gamma \frac{LWC}{N_d} \qquad (8)$$

Where $\gamma$ is a parameter that depends on the DSD shape and droplet size. It can be written as a function of $\varepsilon$, defined as the ratio between the DSD standard deviation and its average, which is much more common in the literature (e.g. Liu and Daum 2002; Tas et al. 2015):

$$\gamma = \frac{\Lambda^2}{(\mu+2)(\mu+1)} = \frac{\Lambda \varepsilon^2}{D_a} \qquad (9)$$

$D_a$ is the mean diameter resulting from the ratio between the 2nd and 1st order moments. By substituting (9) into (8), applying the natural logarithm and the partial derivative to $lnX_i$ (as in Equation 1), it is possible to write:

$$\frac{\partial lnN_d}{\partial lnX_i} = \frac{\partial ln\Lambda}{\partial lnX_i} + 2\frac{\partial ln\varepsilon}{\partial lnX_i} + \frac{\partial lnLWC}{\partial lnX_i} - 2\frac{\partial lnD_{eff}}{\partial lnX_i} \qquad (10)$$

which is an explicit representation of the relation between the sensitivities. Note that $\frac{\partial lnD_{eff}}{\partial lnX_i} = \frac{\partial lnD_a}{\partial lnX_i}$ because of the similarities in the equations of both diameters. The first two terms in the right-hand side of Equation 10 represent the DSD shape, where $\Lambda$ is related to the curvature of the Gamma curve and $\varepsilon$ is the relative dispersion around the DSD mean geometric diameter. Lower (higher) $\Lambda$ and higher (lower) $\varepsilon$ are associated to broader (narrower) DSDs. Equation 10 shows that, in order to compare the sensitivities in $N_d$, $D_{eff}$, and $LWC$, the DSD shape has to be taken into account.

Several aspects of the aerosol-cloud-interaction physics can be illustrated by Equation 10. The Twomey effect states that clouds subject to high aerosol concentrations have enhanced albedo because of the more numerous droplets with increasing aerosol loading (Twomey,1974). This effect is defined when comparing clouds with the same $LWC$. Translating it into Equation 10 (with $X_i = N_a$), it means the $LWC$ derivative is neglected, which defines a relation between droplet concentration, effective diameter, and DSD shape. By comparing to the expression $\overline{S_{D_{eff}}(N_a)} = -\frac{1}{3}\overline{S_{N_d}(N_a)}$ often found in the literature, we can conclude that the sensitivity in $N_d$ is offset by DSD shape alteration. In other words, two thirds of the $N_d$ sensitivity is allocated into DSD narrowing or broadening, while the remainder is effectively altering $D_{eff}$.

The effects of enhanced aerosol concentrations on the DSD shape is of great interest to the climate change community, given that it contributes to the aerosol indirect effect. Liu and Daum (2002) report that pollution, besides lowering droplet size, tends to broaden the DSDs, which would result in weaker cooling forcing compared to previous calculations. They show that the




previous estimations of the aerosol indirect effect considered a fixed $\varepsilon$, possibly overestimating the cooling forcing. Recently, Xie et al. (2017) reports improved model comparisons with satellites when better estimating the relative dispersion. Therefore, it is important to understand the relation between $\varepsilon$ (and $\Lambda$) not only to aerosols, but also to updraft speed and height above cloud base. The overall averages presented in Table 3 show that the DSD curvature ($\Lambda$) is sensitive to $N_a$ and $H$, but the values

are rather small for $\varepsilon$. This results from the not-so-simple relation between DSD shape and $N_a$, $w$ and $H$. Figure 3 shows the variations of the sensitivities of $\Lambda$ and $\varepsilon$ with $N_a$ and $H$ (no significant variations were found for $w$), where it is clear that the overall averages in Table 3 must be analyzed with caution for DSD shape. The $\varepsilon$ sensitivities have significantly different behavior for clean and polluted clouds and also change sign along $H$. Both features result in a low overall average as presented in Table 3, but this does not mean that the $\varepsilon$ sensitivity is negligible. Instead, a more detailed analysis should be considered.

The sensitivities in $\Lambda$ and $\varepsilon$ usually have opposite signs, given their relation to DSD shape – broader DSDs tend to have higher $\varepsilon$ but lower $\Lambda$. Nevertheless, their relation with $N_a$ and $H$ are conceptually similar and illustrate interesting processes. Figure 3a shows that the DSD shape variation with altitude is much more pronounced in cleaner clouds, which is a result of a strong collision-coalescence process. The higher the aerosol concentration, the lower is the sensitivity of $\varepsilon$ to $H$. For the most polluted clouds measured by HALO, the relative dispersion parameter is almost insensible to $H$, meaning that it does not change much

as the cloud grows. There is, however, still some effect on the DSD curvature, making the summation of the first 2 terms in the right-hand side of Equation 10 non-negative in this case (see solid blue line in Figure 3a). For the sensitivities of $\Lambda$ and $\varepsilon$ to $w$, the same summation (dashed line in Figure 3a) is basically null, meaning that these two terms have no contribution in Equation 10. Nevertheless, stronger updrafts tend to produce narrower DSDs in the maritime clouds where the aerosol population is limited in terms of number concentration and particle type/chemistry.

The patterns along $H$ of the DSD shape sensitivities (Figure 3b) pose an interesting question for the parameterization of the aerosol indirect effect in Amazonian clouds. There are significant changes in $\varepsilon$ tendencies as the clouds evolve. Note that aerosols induce broader DSDs up to $H \sim 500$ m, but the opposite happens above that point. In fact, for our higher altitude bin ($2637$ m $< H \leq 4156$ m), the average $\varepsilon$ is lowest for the most polluted clouds ($= 0.28$, while clouds over the forest and Atlantic Ocean show values of $0.32$ and $0.42$, respectively). In other words, the effect of broader DSDs under polluted conditions may

not directly apply for convective clouds over the Amazon, where growth processes in the cloud can significantly change this pattern. This highlights the need to take cloud evolution into account and there is no direct relation between aerosols and cloud relative dispersion in the warm phase of Amazonian clouds. For satellite retrievals, where integrated quantities are of likely interest, the relative dispersion will depend not only on the aerosol concentration but also on cloud depth and lifecycle stage. Regarding the sensitivities to $w$, Figure 3 and Tables 2 and 3 show that updraft speed has little impact on DSD shape or droplet

size. The result in terms of Equation 10 is the equality between the sensitivities in $N_d$ and $LWC$, which is generally the case when we compare the averages shown in Tables 2 and 3. In other words, the updraft effect is limited to increases in the droplet concentration and water content, modulating both $N_d$ and $LWC$ in the same proportion. Overall, the observations shown here should help understand which cloud properties are affected by aerosols, cloud evolution and thermodynamic conditions. The



latter was found to be associated to bulk water contents in the clouds, while the overall shape of the DSDs are determined by the aerosol condition during cloud formation and the subsequent evolution.

## 4 Concluding remarks

The ACRIDICON-CHUVA campaign and the capabilities of the HALO aircraft allowed, for the first time, analyzing the sensitivities of Amazon tropical convective clouds to aerosol number concentrations and updraft speed while also considering cloud evolution. The sensitivity formulation identified that aerosol number concentrations play a primary role in the formation of the warm phase of convective clouds, determining not only droplet concentration but also diameter and overall DSD shape. On the other hand, the thermodynamic conditions, as represented by the updraft intensity, affect primarily DSD bulk properties such as water content and droplet concentration. We have shown that the altitude above cloud base is critical when analyzing aerosol and updraft impacts on clouds, given that the DSD properties evolve with further processing in the system.

We showed that an increase of 100% in aerosol concentration results in an 84% increase in droplet number concentration on average, while the same relative increase in updraft wind speed results in only 43% change. Regarding mean droplet size, we found it to be effectively independent of the updraft speed. Roughly, the effective droplet diameter decreases 25% when aerosol concentration doubles. The comparison between the aerosol and the thermodynamic effects shows that the aerosol concentration is the primary driver for DSD, whereas the updrafts mainly affect droplet number concentration and liquid water content. During cloud evolution, droplet number concentration is depleted while the diameter sensitivity to the growth processes is quantitatively similar to the aerosol effect. Additionally, the aerosol effect on DSD shape inverts in sign with altitude, favoring broader droplet distributions close to cloud base but narrower higher in the clouds. This highlights the importance of differentiating the analysis by altitude above cloud base, which is an appropriate proxy for DSD lifetime for our measurements.

The results presented here can potentially be used to derive new parameterizations in numerical models. They pointed out the specific roles of aerosol particles, updraft speed, and cloud evolution on warm-phase microphysical properties, which can help evaluate the ability of numerical models to reproduce tropical convective clouds.

*Acknowledgements:* The ACRIDICON-CHUVA campaign was supported by the Max Planck Society (MPG), the German Science Foundation (DFG Priority Program SPP 1294), the German Aerospace Center (DLR), the FAPESP (Sao Paulo Research Foundation) grants 2009/15235-8 and 2013/05014-0), and a wide range of other institutional partners. It was carried out in collaboration with the USA–Brazilian atmosphere research project GoAmazon2014/5, including numerous institutional partners. We would like to thank the Instituto Nacional de Pesquisas da Amazonia (INPA) for local logistic help prior, during and after the campaign. Thanks also to the Brazilian Space Agency (AEB: Agencia Espacial Brasileira) responsible for the program of cooperation (CNPq license 00254/2013–9 of the Brazilian National Council for Scientific and Technological Development). The contribution of Dr. Rosenfeld was supported by project BACCHUS European Commission FP7-603445.



Micael A. Cecchini was funded by FAPESP grant 2014/08615-7. The entire ACRIDICON-CHUVA project team is gratefully acknowledged for collaboration and support. The data used in this study can be found at http://www.halo.dlr.de/halo-db/.

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

**Figure captions**

**Figure 1:** Locations where cloud profiles have been collected for different HALO flights. Clouds formed over southern Amazonia and in the Manaus region are subject to higher aerosol loadings due to the presence of the deforestation arc and urban emissions. Clouds formed over the northern and northwestern Amazon are driven by background conditions with low aerosol concentration. During the GoAmazon2014/5 IOP2, maritime clouds were also profiled on the Atlantic coast.

**Figure 2:** Droplet size distributions as function of altitude above cloud base, aerosol particle number concentration, and vertical wind speed, $W$. Four 1000-m-thick layers are considered in the vertical, where the legends in the graphs show the respective upper limit of each one. Solid lines represent averaged DSDs for -1 m s$^{-1}$ ≤ $W$ ≤ 1 m s$^{-1}$, i.e., for relatively neutral vertical movements. Dashed lines represent averaged DSDs for the updraft regions where $W$ > 1 m s$^{-1}$, and dot-dashed lines represent the downdrafts ($W$ < -1 m s$^{-1}$).

**Figure 3:** Variations of the sensitivities of $\Lambda$ and $\varepsilon$ with a) $N_a$ and b) $H$. Note that the sensitivities of $\varepsilon$ are multiplied by 2 in order to be consistent with Equation 10. The curves are averaged over all values of the third dependent variable. For instance, the curve $S_\Lambda(w)$ in a) is averaged over all H values. Blue curves represent the sum of the sensitivities of $\Lambda$ and $\varepsilon$, equivalent to the first two terms in the right-hand side of Equation 10.





**Table captions**

**Table 1:** $N_a$ and CCN at cloud base for each flight considered in this study. *CCN concentrations for flight AC20 showed pronounced scaling with S. The value shown is for the measurements where S>0.52%. This value is closer to the maximum droplet concentration measured at the base of the clouds (=1422 cm⁻³).

5 **Table 2:** $N_d$ and $D_{eff}$ averaged sensitivities to $N_a$, $w$, and $H$. Standard deviations are also shown. R² values are averages of the individual fits. The total variations for $N_a$, $w$, and $H$ are 500 cm⁻³ to 4500 cm⁻³, 0 m s⁻¹ to 8 m s⁻¹ and 0 m to 4156 m, respectively. Intervals grows logarithmically (or close to) for $w$ and $H$.

**Table 3:** Same as Table 2, but for the sensitivities in $\Lambda$, $\varepsilon$, and $LWC$.

**Tables**

10 **Table 1:** $N_a$ and CCN at cloud base for each flight considered in this study. *CCN concentrations for flight AC20 showed pronounced scaling with S. The value shown is for the measurements where S>0.52%. This value is closer to the maximum droplet concentration measured at the base of the clouds (=1422 cm⁻³).

| Flight | $N_a$ (cm⁻³) | CCN (cm⁻³) | S (%) |
|--------|--------------|------------|-------|
| AC19 | 465 | 119 | 0.52 |
| AC18 | 744 | 408 | 0.50 |
| AC9 | 821 | 372 | 0.51 |
| AC20 | 2331 | 1155* | 0.55 |
| AC7 | 2498 | 1579 | 0.50 |
| AC11 | 2691 | 1297 | 0.49 |
| AC12 | 3057 | 2017 | 0.44 |
| AC13 | 4093 | 2263 | 0.44 |



**Table 2:** $N_d$ and $D_{eff}$ averaged sensitivities to $N_a$, $w$, and $H$. Standard deviations are also shown. $R^2$ values are averages of the individual fits. The total variations for $N_a$, $w$, and $H$ are 500 cm$^{-3}$ to 4500 cm$^{-3}$, 0 m s$^{-1}$ to 8 m s$^{-1}$ and 0 m to 4156 m, respectively. Intervals grows logarithmically (or close to) for $w$ and $H$.

|  | $\overline{S_{N_d}}$ | $\overline{S_{D_{eff}}}$ |
|---|---|---|
| $N_a$ | $\mathbf{0.84} \pm 0.21$ | $\mathbf{-0.25} \pm 0.074$ |
|  | $R^2 = 0.91$ | $R^2 = 0.89$ |
| $w$ | $\mathbf{0.43} \pm 0.28$ | $\mathbf{0.028} \pm 0.058$ |
|  | $R^2 = 0.81$ | $R^2 = 0.46$ |
| $H$ | $\mathbf{-0.13} \pm 0.16$ | $\mathbf{0.28} \pm 0.058$ |
|  | $R^2 = 0.38$ | $R^2 = 0.93$ |

**Table 3:** Same as Table 2, but for the sensitivities in $\Lambda$, $\varepsilon$, and $LWC$.

|  | $\overline{S_\Lambda}$ | $\overline{S_\varepsilon}$ | $\overline{S_{LWC}}$ |
|---|---|---|---|
| $N_a$ | $\mathbf{0.23} \pm 0.34$ | $\mathbf{-0.015} \pm 0.16$ | $\mathbf{0.078} \pm 0.34$ |
|  | $R^2 = 0.64$ | $R^2 = 0.54$ | $R^2 = 0.34$ |
| $w$ | $\mathbf{0.046} \pm 0.17$ | $\mathbf{0.039} \pm 0.094$ | $\mathbf{0.49} \pm 0.34$ |
|  | $R^2 = 0.49$ | $R^2 = 0.46$ | $R^2 = 0.77$ |
| $H$ | $\mathbf{-0.43} \pm 0.32$ | $\mathbf{0.094} \pm 0.16$ | $\mathbf{0.67} \pm 0.21$ |
|  | $R^2 = 0.64$ | $R^2 = 0.42$ | $R^2 = 0.76$ |





**Figures**

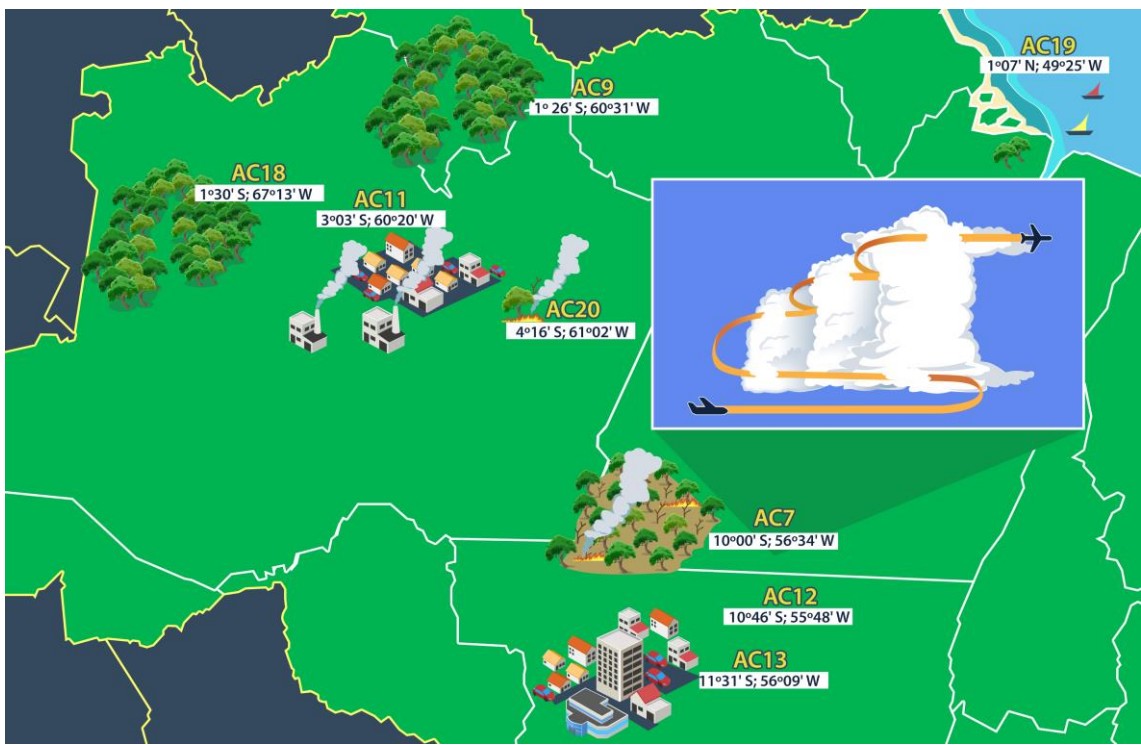

**Figure 1:** Locations where cloud profiles have been collected for different HALO flights. Clouds formed over southern Amazonia and in the Manaus region are subject to higher aerosol loadings due to the presence of the deforestation arc and

5 urban emissions. Clouds formed over the northern and northwestern Amazon are driven by background conditions with low aerosol concentration. During the GoAmazon2014/5 IOP2, maritime clouds were also profiled on the Atlantic coast.





**Figure 2:** Droplet size distributions as function of altitude above cloud base, aerosol particle number concentration, and vertical wind speed, $W$. Four 1000-m-thick layers are considered in the vertical, where the legends in the graphs show the respective upper limit of each one. Solid lines represent averaged DSDs for -1 m s$^{-1}$ $\leq W \leq$ 1 m s$^{-1}$, i.e., for relatively neutral vertical movements. Dashed lines represent averaged DSDs for the updraft regions where $W > 1$ m s$^{-1}$, and dot-dashed lines represent the downdrafts ($W < -1$ m s$^{-1}$).




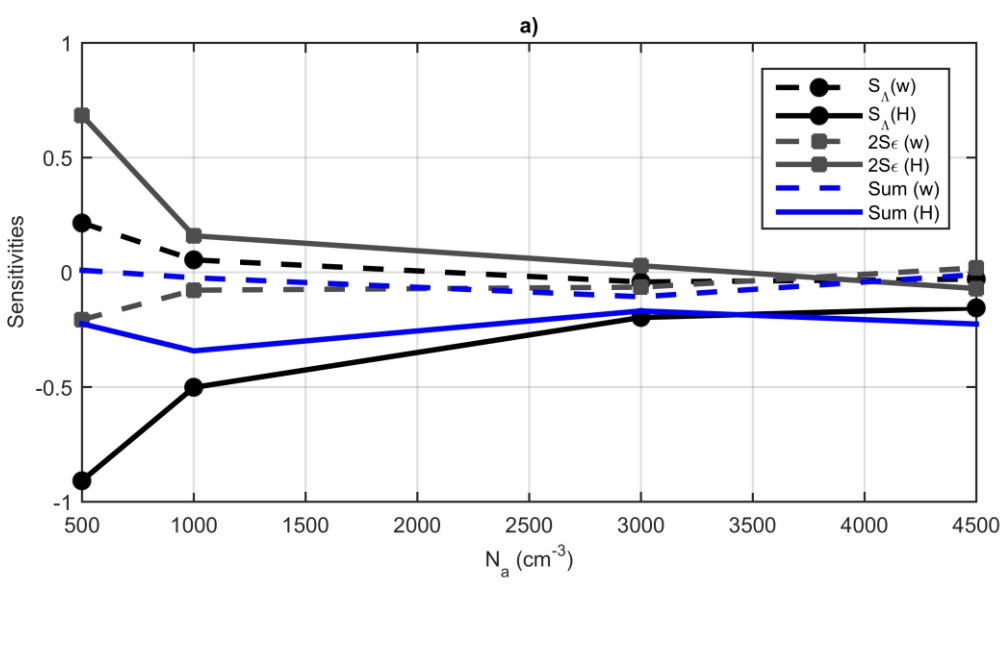

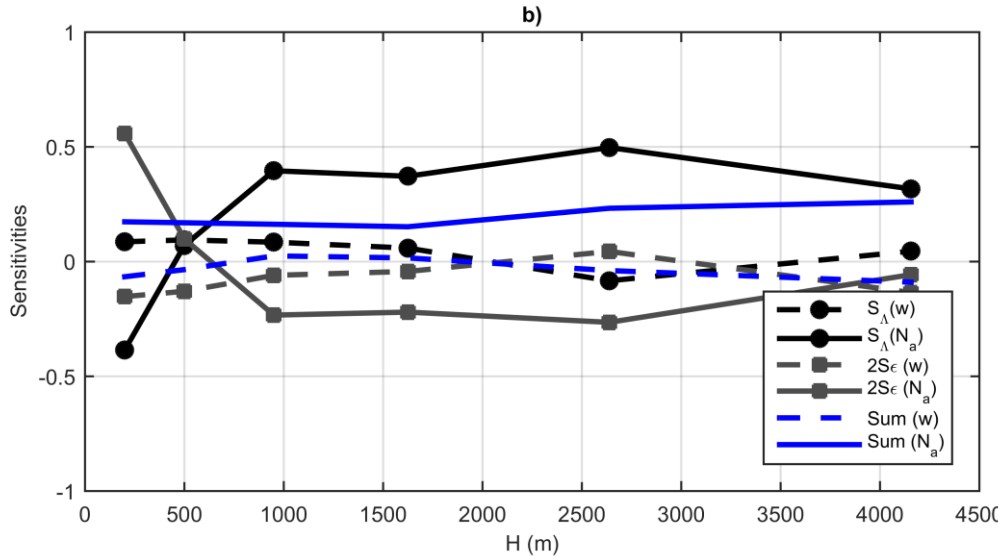

**Figure 3:** Variations of the sensitivities of $\Lambda$ and $\varepsilon$ with a) $N_a$ and b) $H$. Note that the sensitivities of $\varepsilon$ are multiplied by 2 in order to be consistent with Equation 10. The curves are averaged over all values of the third dependent variable. For instance, the curve $S_\Lambda(w)$ in a) is averaged over all H values. Blue curves represent the sum of the sensitivities of $\Lambda$ and $\varepsilon$, equivalent to the first two terms in the right-hand side of Equation 10.