# Peer review of "Sensitivities of Amazonian clouds to aerosols and updraft speed"

_Atmospheric Chemistry and Physics, 2017_

## Referee Comment (RC1) · Anonymous Referee #1 · 18 Mar 2017

Review of Sensitivity of Amazon clouds to aerosols and updraft speed, by Cecchini et al.

This manuscript uses aircraft in situ observations of developing cumulus clouds over the Amazon to explore the connection between control variables (here the concentration Na of aerosol particles entering clouds from below, the height h above cloud base, and the updraft speed, w) and cloud microphysical properties (cloud droplet concentration, effective radius and spectral shape) at levels above cloud base. The connections are determined by an approach that attempts to isolate the effect of a particular control variable by holding the other control variables fixed (partial derivative approach). This is carried out in practice by binning the data into three dimensional bins of Na, h and w. The findings indicate that the primary controls on effective radius are h and Na, and that the primary control on Nd is Na with w also positively influencing Nd. LWC is

mainly controlled by h and w. These results seem to make physical sense. There are new findings about factors controlling the drop size shape that will be of interest to the community. The manuscript is relevant to the cloud community, although it is not clear what modelers would do with the results. I find the manuscript suitable for publication in Atmospheric Chemistry and Physics, and offer some suggestions for revision.

SPECIFIC COMMENTS:

1. The finding that Nd is strongly correlated with Na differs from studies in shallow broken cumulus (e.g. Vogelmann et al. 2011, BAMS, Fig. 10) where Nd does not appear to be strongly correlated with Na and LWP decreases with Na. Some discussion of contrasts with prior work would put this work into the context of previous work.

2. The justification for why Na is used rather than CCN does not make sense to me. The CCN measurements include supersaturations up to 0.55%, so why not just choose a fixed supersaturation (interpolate if needed) and use CCN. The authors should use both CCN and Na and compare the results.

3. The authors should investigate the impacts of the rather large bin sizes they need to compute the partial derivatives on the values of the derivatives.

4. P1, Line 32. I disagree that height above cloud base is a good proxy for time in cloud. A much better estimate would be h/w, which actually has the units of time and would be the exact time in cloud if w is constant with height. The authors should re-evaluate their conclusions in the light of this error.

5. McFiggans et al. (2006, ACP) have a good review paper exploring factors controlling cloud microphysics. The results here could be put into the context of the findings in that paper.

6. P5, line 27. What are linear and angular coefficients?

7. Was there any correlation between w and Na? How might this change the results?

8. P11, Line 24-26. I couldn't follow this argument at all. Others will probably have difficulty with it. Sensitivity in Nd to what?

9. P12, Line 14. Insensitive rather than insensible.

---

## Referee Comment (RC2) · Anonymous Referee #2 · 12 May 2017

This paper by Cecchini et al describes the use of the German HALO aircraft, with a very comprehensive payload for studying atmospheric physics and chemistry, and powerful performance characteristics in range and altitude, to study clouds forming over Amazonia. The results are important because Amazonia is an understudied region which is capable of having a profound influence on the Earth's climate. They appear to show a clear dominance of the influence of particle number concentration in air entering the cumulus clouds on their further development. Other factors such as updraft speed have less influence as does height above cloud base.

The paper is well written in the most part and is acceptable for publication subject to dealing with the comments below.

1. It would be much easier to read this paper if it contained a table defining the many

physical quantities in the equations, and in the diagrams. Ideally this should be Table 1.

2. Page 7, line 22: The text refers to supplementary material in the form of figures S1-S4. These are not shown in the manuscript. Are these shown in an appendix somewhere?

3. The real physical significance of Figure 2 could be better explained by describing the shape of the lines drawn in the figure as the droplet size 'D' increases particularly the significance of the inflexions. Reasons could be given for why droplet size continues to increase with altitude. To some extent this is dealt with in the concluding remarks but for clarity should be included when the figure is described in detail.

4. The axis labelling and figure caption shown in Figure 2 needs improvement. In particular, is the vertical axis na – is the same quantity as shown in Table 1. Some linkage between the numbers referred to in Table 1 and Figure 2 would be helpful.

5. It appears from Table 1 that there is a significant difference in particle number or CCN, and possibly chemical composition, between maritime and continental cumulus clouds. Are there obvious differences in cloud appearance and shape associated with the differing input parameters? This would be suggested from their conclusions regarding the importance of particle number on cloud development.

6. Is there any information on the chemical composition of particles entering the different clouds, and in particular regarding the contrast with flight AC19 and the rest? Figure 2 suggests there should be.

7. Is there a diagram showing the total aircraft instrumentation package and its capabilities in the series dealing with the overall experiment in Amazonia? Perhaps this is described elsewhere and if so should be referenced. Perhaps the Wendisch et al 2016 paper covers this.

---

## Author Comment (AC1) · 31 May 2017

**COMMENTS RECEIVED FROM REVIEWER #1**

**General comments**

This manuscript uses aircraft in situ observations of developing cumulus clouds over the Amazon to explore the connection between control variables (here the concentration $N_a$ of aerosol particles entering clouds from below, the height $H$ above cloud base, and the updraft speed, $w$) and cloud microphysical properties (cloud droplet concentration, effective radius and spectral shape) at levels above cloud base. The connections are determined by an approach that attempts to isolate the effect of a particular control variable by holding the other control variables fixed (partial derivative approach). This is carried out in practice by binning the data into three dimensional bins of $N_a$, $H$ and $w$. The findings indicate that the primary controls on effective radius are $H$ and $N_a$, and that the primary control on $N_d$ is $N_a$ with $w$ also positively influencing $N_d$. $LWC$ is mainly controlled by $H$ and $w$. These results seem to make physical sense. There are new findings about factors controlling the drop size shape that will be of interest to the community. The manuscript is relevant to the cloud community, although it is not clear what modelers would do with the results. I find the manuscript suitable for publication in Atmospheric Chemistry and Physics, and offer some suggestions for revision.

**Authors answers**

We would like to express our gratitude for the anonymous Reviewer #1 for taking the time to review this manuscript. Your suggestions are invaluable and very helpful. We will try our best to address all of them.

Regarding your concerns on how the results can be of use to modelers, here are our suggestions (which we will try to make clearer in the text). Our sensitivity results (i.e. Tables 2-3) can be used for direct comparisons with model results. Tropical clouds in general, or Amazonian clouds in our case, are still poorly represented by models. One of the common issues is the representation of the precipitation daily cycle, where rainfall tends to occur earlier in models compared to observations. We believe one reason for that is the misrepresentation of the DSDs that can lead to artificially high efficiency in rain formation. Therefore, model runs can be performed in order to assess the factors that control DSD formation and comparisons can be made with our results as benchmark. The analysis of the $\varepsilon$ and $\Lambda$ parameters can be especially useful in that regard. In that case, however, it should be beneficial to consider the more detailed results shown in Figure 3. This figure shows that aerosols can induce DSD broadening only close to cloud base, preferably under high $w$ conditions. Higher in the clouds, increased aerosol loading leads to DSD

narrowing. The variability of DSD shape with altitude is pronounced in cleaner clouds, decreasing with increasing $N_a$. The updraft effect on DSD shape is secondary, but enhanced updrafts may lead to narrower DSD in clean clouds given the limited aerosol availability. Good models should be able to reproduce such details in order to generate better forecasts. Therefore, we believe our results can be of use in that direction, by providing specificities of Amazonian clouds that models should aim to reproduce.

The last paragraph (right before the acknowledgements) was changed to reflect this feedback:

"The results presented here can potentially be used to validate and derive new parameterizations in numerical models, which usually fail to correctly represent Amazonian convective clouds. One common issue of the models is the representation of the precipitation daily cycle, where the modelled rainfall tends to occur earlier than in the observations. One possible reason for that is the misrepresentation of the cloud DSDs that can lead to artificially high efficiency in rain formation. Therefore, model runs can be performed in order to assess the factors that control DSD formation and comparisons can be made with our results as benchmark. The analysis of the $\varepsilon$ and $\Lambda$ parameters can be especially useful in that regard. The results presented here detail several aspects of the Amazonian clouds and their relation to aerosol and thermodynamic conditions. For instance, it was shown that aerosols can induce DSD broadening only close to cloud base, preferably under high $w$ conditions. Higher in the clouds, increased aerosol loading leads to DSD narrowing. Additionally, DSD broadening with altitude is pronounced only in clean clouds, where the collection processes are efficient. The result is growing $\varepsilon$ with altitude, while this parameter remains relatively constant with $H$ in polluted clouds. Good models should be able to reproduce such details in order to generate better forecasts. Therefore, we believe the results presented here can be of use in that direction, by providing specificities of Amazonian clouds that models should aim to reproduce".

**Specific comments from Reviewer #1**

1.

   a. **(Question)** The finding that $N_d$ is strongly correlated with $N_a$ differs from studies in shallow broken cumulus (e.g. Vogelmann et al. 2011, BAMS, Fig. 10) where $N_d$ does not appear to be strongly correlated with $N_a$ and *LWP* decreases with $N_a$. Some discussion of contrasts with prior work would put this work into the context of previous work.

   b. **(Answer)** Thank you for the reference. We specifically address comparison with other studies between P8 Line 29 and P9 Line 11. Nevertheless, your reference prompts an interesting question regarding the sensitivity results and meteorological conditions. We added the following sentences starting at P9 Line 17:

   "The meteorological and cloud morphology conditions in the Amazon also seem to enable the high sensitivity values found. A previous study by Vogelmann et al. (2012) found relatively invariant $N_d$ as function of $N_a$. Beyond instrumental and methodological differences, this study also focused on shallow (200 m to 500 m thickness), broken clouds with weak updrafts over Oklahoma. This type of cloud favors the entrainment mixing feedback, where polluted clouds tend to have lower *LWC* because of enhanced droplet evaporation. The differences between the results shown here and the study of Vogelmann et al. (2012) suggest that the entrainment mixing process is not dominant over the Amazon. Possible reasons include abundant water vapor, thicker clouds, stronger convection and updrafts, and low vertical wind shear. High humidity of the surrounding air induces weaker *LWC* and $N_d$ depletion by the entrainment mixing process (see, for instance, Korolev et al. 2016) because of slower evaporation. Stroger convection induces deeper clouds that have a relatively low area-to-volume ratio as compared to the clouds reported in Vogelmann et al. (2012). Therefore, the entrainment at cloud edges are not as dominant. Low area-to-volume ratios are also favored by the weak vertical wind shear typical of tropical regions. This mechanism was studied in Fan et al. (2009) that concluded that convection invigoration is favored under low vertical wind shear conditions, while the opposite happens with high vertical wind shear".

2.

a. **(Question)** The justification for why $N_a$ is used rather than CCN does not make sense to me. The CCN measurements include supersaturations up to 0.55%, so why not just choose a fixed supersaturation (interpolate if needed) and use CCN. The authors should use both CCN and $N_a$ and compare the results.

b. **(Answer)** We understand the concern because, at first glance, CCN concentrations seems to be the way to go. However, it is known that clean and polluted clouds can present different supersaturation conditions, making the use of a constant-supersaturation CCN measurement not representative of the cloud variability we observed. In this case, it is hard to obtain a common benchmark to make comparisons. Nevertheless, the averaged CCN and $N_a$ below clouds proved to be linearly correlated and $N_a$ does not change with supersaturation. Calculating partial derivatives to $N_a$ or to the proportional CCN would not change the results. Added the following text at P5 Line 29:

"It is known that polluted clouds tend to have lower supersaturations given the enhanced condensation. Therefore, the use of constant-supersaturation CCN concentrations does not provide a common benchmark between the clouds probed here. Conversely, it is difficult to obtain the supersaturation within the clouds and the consequent CCN concentration modulation. In that regard, $N_a$ proved to be most adequate for providing a framework to compare polluted and clean clouds".

3.

a. **(Question)** The authors should investigate the impacts of the rather large bin sizes they need to compute the partial derivatives on the values of the derivatives.

b. **(Answer)** This is a tricky process. We believe the bin sizes are as large as they can possibly be. Therefore, the only option would be to assess the effects of narrower bins on the results. However, even if there are differences between the narrow-bin and broad-bin cases, it is difficult to assess the significance because of the lower statistical confidence of the former case. Nonetheless, we performed a test by increasing the number of $w$ and $H$ bins. We maintained the same overall interval (0 ms$^{-1}$ < $w$ < 8 ms$^{-1}$ and 0 km < $H$ < ~4 km) and added one $w$ bin and three $H$ bins. By recalculating the sensitivities and comparing to the results shown in Tables 2-3, we obtained a maximum difference of 0.064. Therefore, narrower bin sizes would not result in significantly different results.

Added the following sentence on the end of Section 2.2: "Different bin configurations were tested and the results proved to be relatively insensitive to the bin number and width".

4.

a. **(Question)** P1, Line 32. I disagree that height above cloud base is a good proxy for time in cloud. A much better estimate would be $H/w$, which actually has the units of time and would be the exact time in cloud if w is constant with height. The authors should re-evaluate their conclusions in the light of this error.

b. **(Answer)** We understand your concern given that $H$ may seem an arbitrary choice. However, we believe there is no real gain by analyzing the results in terms of $H/w$ instead of $H$ only. If we were to directly estimate the cloud lifetime from the $H/w$ ratio, we would have to prescribe the $w$ vertical profile. Even though we have some measurements in the clouds, this would be very difficult given that $w$ also varies greatly in the horizontal direction (contrast between cloud core and edges). On the other hand, if we consider a constant $w$, then $H$ would be directly proportional to $H/w$, meaning that $H$ is an effective proxy for cloud lifetime as calculated from this ratio. Taking into account the measurement strategy and that we are measuring almost exclusively growing convective elements, the choice of $H$ seems to be the most direct one. We made this point clearer in the text by adding the following discussion in the 4[th] paragraph of Section 2.2: "It could be argued that the ratio $H/w$ would be a more direct estimate of the cloud lifetime, given that it is the time that it took for the cloud to reach $H$. However, this approach would need prescribed $w$ profiles below each measurement, which is not feasible in this study given that different clouds can be measured in the same profiling mission. Additionally, there is high $w$ variability horizontally between the clouds edges and cores, adding extra complexity. Therefore, we will use $H$ as the proxy for cloud evolution even though it does not represent cloud lifetime directly (i.e. does not have units of time). The profiling strategy of measuring growing convective clouds favors this interpretation".

5.

a. **(Question)** McFiggans et al. (2006, ACP) have a good review paper exploring factors controlling cloud microphysics. The results here could be put into the context of the findings in that paper.

b. **(Answer)** We cite this paper in the first sentence of Section 2.2. We do provide discussions considering the overall context of the sensitivity approach. We present the papers that performed sensitivity calculations, McFiggans et al. (2006) being one of them, and highlight their limitations and how we aim to improve such analysis. Therefore, we believe we are already putting our results into perspective.

6.

a. **(Question)** P5, line 27. What are linear and angular coefficients?

b. **(Answer)** Those are the parameters we obtained from the regression curve between $N_a$ and *CCN*. The sentence has been changed slightly following this feedback.

7.

a. **(Question)** Was there any correlation between $w$ and $N_a$? How might this change the results?

b. **(Answer)** Firstly, we consider only a constant value for $N_a$ for each flight (the averaged measurement around the cloud base altitude). Therefore, for cloud base, there would be several $w$ values for only one $N_a$ and we cannot calculate the correlation between them. Nonetheless, any correlations between $w$ and $N_a$ (or $H$) are minimized by the binning procedure. When we calculate the sensitivities by fixing two bins and varying the third, the interdependences between $N_a$, $w$, and $H$ are eliminated.

8.

a. **(Question)** P11, Line 24-26. I couldn't follow this argument at all. Others will probably have difficulty with it. Sensitivity in $N_d$ to what?

b. **(Answer)** It is sensitivity of $N_d$ to $N_a$, as can be inferred from context. Changed the sentence slightly to: "By comparing to the expression $\overline{S_{D_{eff}}(N_a)} = -\frac{1}{3}\overline{S_{N_d}(N_a)}$ often found in the literature, we can conclude that the value of the sensitivity of $N_d$ to $N_a$ is offset by some effect on DSD shape. In other words, two thirds of the $N_d$ sensitivity is allocated into DSD narrowing or broadening, while the remainder is effectively altering $D_{eff}$".

9.

a. **(Question)** P12, Line 14. Insensitive rather than insensible.

b. **(Answer)** Thanks.

---

## Author Comment (AC2) · 31 May 2017

**COMMENTS RECEIVED FROM REVIEWER #2**

**General comments**

This paper by Cecchini et al describes the use of the German HALO aircraft, with a very comprehensive payload for studying atmospheric physics and chemistry, and powerful performance characteristics in range and altitude, to study clouds forming over Amazonia. The results are important because Amazonia is an understudied region which is capable of having a profound influence on the Earth's climate. They appear to show a clear dominance of the influence of particle number concentration in air entering the cumulus clouds on their further development. Other factors such as updraft speed have less influence as does height above cloud base. The paper is well written in the most part and is acceptable for publication subject to dealing with the comments below.

**Authors answer**

We would like to express our gratitude for the anonymous Reviewer #2 for taking the time to review this manuscript. Your suggestions are addressed in detail in this document, please see the list below.

**Specific comments from Reviewer #2**

1.
    a. **(Question)** It would be much easier to read this paper if it contained a table defining the many physical quantities in the equations, and in the diagrams. Ideally this should be Table 1.
    b. **(Answer)** Added the requested table as Table 1.

2.
    a. **(Question)** Page 7, line 22: The text refers to supplementary material in the form of figures S1-S4. These are not shown in the manuscript. Are these shown in an appendix somewhere?
    b. **(Answer)** Figures S1-S4 (as well as Tables S1-S18) are part of the supplement material provided. It can be found in this link: http://www.atmos-chem-phys-discuss.net/acp-2017-89/acp-2017-89-supplement.pdf.

3.

    a. **(Question)** The real physical significance of Figure 2 could be better explained by describing the shape of the lines drawn in the figure as the droplet size 'D' increases particularly the significance of the inflexions. Reasons could be given for why droplet size continues to increase with altitude. To some extent this is dealt with in the concluding remarks but for clarity should be included when the figure is described in detail.

    b. **(Answer)** The patterns and inflexions in the curves can be explained by the following concepts. Basically, the balance between the water vapor condensation/evaporation and the collision-coalescence process will ultimately define the DSD curve. Initially the droplets grow by condensation, but latter in the cloud lifetime they also go through the collision-coalescence process. This process occurs when the bigger droplets collide with and collect the smaller droplets. This results in enhanced concentrations of big droplets in detriment of smaller ones. Additionally, there can also occur droplet breakup during the collisions. Both the droplet breakup and the collision-coalescence result in wider DSDs – that is why they are wider higher in the clouds. It is usually considered that measurements higher in the clouds are relative to later stages in the cloud lifetime (because clouds grow from cloud base and upwards). Therefore, the droplet growth with altitude is a result of continuous condensational/collection growth. We added the following sentences to the first paragraph in Section 3.1: "In general, all profiles show droplet growth with altitude as they continually go through the condensational and collision-coalescence processes. The enhanced DSD widening with altitude presented in Figures 2a,b suggest relative predominance of collision-coalescence. Those observations also support the choice of *H* as proxy for cloud lifetime".

4.

    a. **(Question)** The axis labelling and figure caption shown in Figure 2 needs improvement. In particular, is the vertical axis na – is the same quantity as shown in Table 1. Some linkage between the numbers referred to in Table 1 and Figure 2 would be helpful.

    b. **(Answer)** The vertical axis in Figure 2 is actually the DSDs. They represent droplet number concentrations in each $cm^{-3}$ of air and in a log-space diameter interval. That is why we represent it as dN/dlogD. We added the following text

in parenthesis to the second sentence in Section 3.1 to make it clearer: "This figure shows DSDs (dN/dlogD in the vertical axis)…".

5.

    a. **(Question)** It appears from Table 1 that there is a significant difference in particle number or CCN, and possibly chemical composition, between maritime and continental cumulus clouds. Are there obvious differences in cloud appearance and shape associated with the differing input parameters? This would be suggested from their conclusions regarding the importance of particle number on cloud development.

    b. **(Answer)** It is important to highlight the differences between cloud macro- and microphysical aspects. The microphysics is concerned with smaller-scale processes such as droplet growth and water phase changes. On the other hand, aspects such as cloud shape, size, volume, and so on are related to the cloud macrophysics. The results presented in this study pertain to the clouds microphysics. That is, the properties of the droplets in small volumes inside the clouds. The overall appearance and shape of the clouds is beyond the scope of this work. That said, maritime and continental clouds can have different visual aspects, resulting not only from the different pollution levels but also from different meteorology. Even though aerosols have a determinant role in the cloud microphysics, their overall (macroscale) aspect will be determined by the meteorological conditions. For instance, different vertical wind shear produce different shapes of clouds, where they are tilted under strong vertical wind shear. Additionally, turbulent processes around the cloud edges can also influence its shape. When there is strong entrainment mixing at the cloud edges, the overall size of the cloud can be reduced because of evaporation. Such aspects are briefly mentioned in the text, but are not the focus of this study.

6.

    a. **(Question)** Is there any information on the chemical composition of particles entering the different clouds, and in particular regarding the contrast with flight AC19 and the rest? Figure 2 suggests there should be.

    b. **(Answer)** Yes, we expect different aerosol chemical composition between the different regions. While the maritime aerosols contain mainly sea-slat, aerosols over the Amazon forest can have a predominance of biological material. Urban or biomass-burning pollution will add inorganic particles to the aerosol population over the forest. The sources and characteristics of the aerosols over

the Amazon were reviewed by Martin et al. (2010) – cited in the paper. We added the following sentence on the end of paragraph 2 in Section 2.2 to make it clearer that we won't focus on aerosol chemical composition: "In this study, we will focus on aerosol number concentrations and their chemical composition will not be addressed".

7.

    a. **(Question)** Is there a diagram showing the total aircraft instrumentation package and its capabilities in the series dealing with the overall experiment in Amazonia? Perhaps this is described elsewhere and if so should be referenced. Perhaps the Wendisch et al 2016 paper covers this.

    b. **(Answer)** Yes, this can be found in Wendisch et al. (2016). This reference contains the overall informations regarding the ACRIDICON-CHUVA campaign.